# Effects of *Chinese yam* Polysaccharides on the Muscle Tissues Development-Related Genes Expression in Breast and Thigh Muscle of Broilers

**DOI:** 10.3390/genes14010006

**Published:** 2022-12-20

**Authors:** Jiahua Deng, Jinzhou Zhang, Yan Jin, Yadi Chang, Mingyan Shi, Zhiguo Miao

**Affiliations:** 1College of Animal Science and Veterinary Medicine, Henan Institute of Science and Technology, Xinxiang 453003, China; 2Life Science College, Luoyang Normal University, Luoyang 471022, China

**Keywords:** *Chinese yam* polysaccharides, broilers, muscle tissue, gene expression, correlations analysis

## Abstract

This experiment was conducted to evaluate the effects of dietary *Chinese yam* polysaccharides (CYP) on myogenic differentiation 1 (*MYOD1*), myogenin (*MYOG*), and myostatin (*MSTN*) mRNA expression of breast and thigh muscle tissues in broilers. A total of 360 (1-day-old, gender-balanced) crossbred broilers chicks with similar body weight (BW) were randomly distributed into four groups, with three replicates in each group and each replicate included 30 broilers. The feeding trial lasted for 48 days. Experimental broilers were fed 0.00 mg/kg basal diet (control group), 250 mg/kg, 500 mg/kg, and 1000 mg/kg CYP, respectively. The results showed that CYP250 and CYP500 groups had higher thigh muscle percentage (TMP) compared to the control group (*p* < 0.05). Meanwhile, the expression of *MYOD1*, *MYOG* mRNA in breast muscle tissues of CYP500 and CYP1000 groups was higher (*p* < 0.05), and the expression of *MSTN* mRNA in thigh muscle of CYP250, CYP500, and CYP1000 groups was lower than that of the control group (*p* < 0.05). In addition, there was no significant difference in the expression of *MYOD1* mRNA in the thigh muscle tissue of each group (*p* > 0.05). Bivariate correlation analysis showed that the expression levels of *MYOD1*, *MYOG*, and *MSTN* mRNA in the thigh muscle tissue of broiler chickens in the CYP500 group were positively correlated with TMP. However, the expression of *MYOG* mRNA in thigh muscle tissue of the CYP1000 group was negatively correlated with TMP. In general, this study indicated that appropriate dietary CYP supplementation influenced the growth and development of thigh muscle tissue in broilers by altering TMP and muscle tissue development-related genes expression. Therefore, CYP could be used as a potential feed additive to promote the development of muscle tissues in broilers.

## 1. Introduction

As the biological resources of natural plants, Chinese herbs are a source of green feed additive with no residue, wide source, and low cost, which contains polysaccharides, organic acids, flavonoids, and other bioactive substances [1]. Chinese herbal polysaccharides (CHPs) are secondary metabolites extracted from Chinese herbal medicines such as the *Chinese yam* (CY) and *Lycium barbarum*. They have a variety of biological functions, such as regulating intestinal microflora, antioxidant and anti-inflammatory responses, anti-tumor, and so on [2,3,4], which can improve the growth performance and immunity of animals [5], therefore, in recent years, polysaccharides of Chinese herbal medicine have become the research focus of many scholars. CY belongs to the dry rhizome of *Dioscorea opposita Thunb* [6], which is widely distributed in the tropical and subtropical regions of China [7]. The 2010 edition of *The Pharmacopoeia of the People’s Republic of China* clearly records that yam, as one of the traditional Chinese herbs, has edible and medicinal value in China for more than 1500 years [8]. It contains polysaccharides, proteins, starches, amino acids, etc., biologically active ingredients [9,10], among which CYP extracted from CY are mainly included in glucose (Glu) and galactose (Gal) [11]. Animal experiments research found that CYP has many functions, such as promoting animal growth, enhancing antioxidation, and immune function. The average daily food intake and average daily body weight gain of the rats fed the containing CYP diet were greater than those of the control group [12]. CYP supplements showed scavenging activity on the 2,2-Diphenyl-1-picrylhydrazyl (DPPH) radical, hydroxyl radical and superoxide anion radical in increasing concentration [13]. Adding CYP to the diet increases the proliferation activity of lymphocytes and cytokine levels in weaned rats [12]. Our previous study also confirmed that CYP has a positive effect on improving immune function of crossbred broilers [14].

Skeletal muscle is one of the important tissues for weight gain in broilers, including breast and thigh muscles [15], with its development regulated by *MYOD1*, *MYOG*, *MSTN*, and other regulatory factors [16,17]. *MYOD1* is a skeletal muscle-specific transcription factor, which can act on activation, proliferation, and differentiation of satellite cells, and plays an important biological regulatory role in fast- and slow-muscle formation [18,19]. *MYOG* is a key regulator of skeletal muscle differentiation and plays an important role in terminal myoblast differentiation [20,21]. *MSTN*, also known as growth differentiation factor 8 (GDF-8), negatively regulates skeletal muscle growth by blocking myoblast proliferation and differentiation to control the number of muscle fibers [22,23]. However, there is no report about the effect of adding CYP on the differential expression of *MYOD1*, *MYOG,* and *MSTN* gene mRNA in broiler breast muscle and thigh muscle tissue. In order to explore the relationship between CYP and the growth and development of muscle tissues in broilers, the purpose of this experiment is to study the effect of adding CYP in the diet on broiler breast muscle percentage (BMP), TMP and gene expression related to muscle development, and the relationship between gene expression and BMP or TMP. These results will help us understand the regulatory function of CYP in the growth and development of muscle tissue and the expression of related genes in broilers.

## 2. Materials and Methods

### 2.1. Experimental Design, Diets and Broilers

All procedures involving broilers feeding were reviewed and approved by the Animal Protection and Utilization Committee of Henan Institute of Science and Technology (No. 2021HIST018, Xinxiang, China). A total of 360 (1-day-old, gender-balanced) crossbred broilers chicks with similar initial BW (39.54 ± 0.51 g) were randomly allotted to 4 groups (Control, CYP_250_, CYP_500_, and CYP_1000_, respectively) with 3 replicates in each group and each replicate included 30 broilers (sex balanced). The control group was fed a basal diet without CYP, and the CYP_250_, CYP_500_, and CYP_1000_ groups received the same basal diets included 250, 500, and 1000 mg/kg CYP, respectively. Feeding trials were divided into early (1–28 days) and late (29–48 days). The basal diet meeting the National Research Council (NRC, 1994) requirements is shown in Table 1. All broilers had free access to food and water during the experiment. Crossbred broilers chicks are raised in appropriate closed cages to control the air temperature. The lamp is turned on continuously for 24 h, and the temperature is kept at 32 ℃ for the first 3 days, then it is gradually cooled until 26 ℃ from days 4 to 21. The CYP in this study were provided by Shanxi Hanna Biotechnology Co., Ltd. (Xian, Shanxi, China) carbohydrate level more than 90%, polysaccharide content exceeds 30.00% (of which monosaccharides include 99.48% Glu and 0.52% Gal).

### 2.2. Slaughter and Samples Collection

On day 48, after fasting for 12 h, 6 crossbred broilers were randomly selected from each group (2 broilers per replicate, gender balance). After the neck was bled and slaughtered, breast and thigh muscle were harvested and weighed respectively, and used for carcass performance determination. The percentage of breast and thigh muscle tissues are calculated as a percentage of the eviscerated weight. Meanwhile, breast and thigh muscle tissue were collected from the same site, preserved in liquid nitrogen, and then stored at −80 °C in a freezer until further analysis.

### 2.3. Messenger RNA Expression Analysis

Total RNA was detached from breast and thigh muscle tissues by employing TRizol reagent (Takara Biomedical Technology Co., Ltd., Beijing, China). After extraction, total RNA utilizing a spectrophotometer (Biodrop µLite, Cambridge, United Kingdom) measured the concentration of RNA within the A260:A280 ratio range. The RNA was reverse transcribed using test kits by Thermo Fisher Scientific to make cDNA. Each sample was measured in triplicate. Relative mRNA expression of *β-actin*, *MYOD1*, *MYOG,* and *MSTN* was determined by the 2^−ΔΔCt^ method.

### 2.4. Primer Design

Based on GenBank (accessed on 18 June 2021 https://www.ncbi.nlm.nih.gov/genbank/) known sequences were designed with Primer 5.0 software *β-actin*, *MYOD1*, *MYOG,* and *MSTN* mRNA primers [24]. All primer sequences were provided by Sangon Biotech Co. (Shanghai, China) and all primer information is shown in Table 2.

### 2.5. Statistical Analysis

All data were presented as mean ± SEM (standard error of the means). Muscles percentage and gene expression data were analyzed using a one-way ANOVA test. Prism 8 (GraphPad Prism, La Jolla, CA, USA) was used to generate graphs. SPSS 26.0 for Windows (IBM Corp., Chicago, IL, USA) was used to perform the Pearson’s correlation coefficient two tail test to evaluate the correlation between muscle development-related genes expression and BMP or TMP. *p* < 0.05 was considered statistically significant.

## 3. Results

### 3.1. BMP and TMP

Effects of dietary CYP on the BMP and TMP in broilers are shown in Figure 1. No differences in BMP were found among all groups (*p* > 0.05). However, compared with the control group, the TMP of broilers in the CYP_250_ and CYP_500_ groups was higher (*p* < 0.05).

### 3.2. Muscle Development-Related Genes Expression

As shown in Figure 2, in breast muscle tissues, CYP_500_ and CYP_1000_ groups had higher *MYOD1* mRNA expression compared to the control group (*p* < 0.05). Meanwhile, compared to the control group, the CYP_500_ and CYP_1000_ groups had higher *MYOG* expression, CYP_250_ and CYP_500_ groups had higher *MSTN* expression (*p* < 0.05).

As shown in Figure 3, there was no difference in the expression of *MYOD1* mRNA in the thigh muscle tissues of each group (*p* > 0.05), while *MYOG* and *MSTN* mRNA expression in CYP_250_, CYP_500_, and CYP_1000_ groups was significantly lower than that in the control group (*p* < 0.05). In addition, *MYOG* and *MSTN* mRNA expression of the CYP_500_ group was higher than that in the CYP_250_ and CYP_1000_ groups (*p* < 0.05).

### 3.3. Correlations between Genes Expression and BMP or TMP

As shown in Table 3, in breast muscle tissue, the *MYOD1*, *MYOG,* and *MSTN* mRNA expression showed no significant correlation with BMP, TMP in each group (*p* > 0.05).

As shown in Table 4, the *MYOD1*, *MYOG,* and *MSTN* mRNA expression was significantly positively correlated with TMP in the CYP_500_ group (*MYOD1*/TMP, content: r = 0.819, *p* = 0.046; *MYOG*/TMP, content: r = 0.912, *p* = 0.011; and *MSTN*/TMP, content: r = 0.851, *p* = 0.032, respectively), while the *MYOG* mRNA expression was negatively correlated with TMP in the CYP_1000_ group (*MYOG*/TMP, content: r = −0.973, *p* = 0.001).

## 4. Discussion

Previous study showed that BMP and TMP are important traits of carcass yield, which are closely related to the increase of broiler growth and carcass yield [25,26]. Many studies have shown that different kinds of CHPs can increase the TMP of poultry. Sun et al. [27] observed that 1000 mg/kg supplemented with *Astragalus membranaceus* polysaccharides (AMP) significantly enhanced the TMP of chickens. Zhang et al. [28] found that 1% supplemented with AMP can significantly increase the TMP in broilers, thus confirming that AMP could enhance the carcass yield. Sun et al. [29] also found that *L. barbarum* polysaccharides (LBP) additives could improve TMP and enhance the muscle tissue development of early chickens. The present study found that dietary appropriate CYP supplementation could significantly increase TMP of broilers. Our study was similar to previous reports showing that CHPs can increase TMP and promote muscle tissues development in broiler chickens.

Numerous evidence demonstrated that *MYOD1*, *MYOG,* and *MSTN* play key roles in the muscle growth and development. *MYOD1* and *MYOG* are members of the myogenic regulatory factors (MRFs) family, and their expression levels are closely related to the growth rate of muscle tissues [30,31]. As an effective growth and differentiation factor, *MSTN* has been shown to be involved in inhibiting muscle precursor cell proliferation, which determines muscle tissue growth and development [32]. In addition, *MSTN* is also involved in the formation of skeletal muscle in broilers [33]. In the current study, CYP_500_ and CYP_1000_ significantly enhanced the mRNA expression of the *MYOD1* and *MYOG* in breast muscle tissues. In addition, a diet which is CYP supplemented significantly downregulated the mRNA expression of the *MSTN* in thigh muscle tissues of broilers. These results indicate that dietary appropriate CYP supplementation has a positive effect on the expression of *MYOD1* and *MYOG* genes related to muscle development in broilers, and can also inhibit the expression of the negative regulator *MSTN*. All of these results suggest that the CYP promotes broiler muscle development by regulating the expression of muscle-related genes, which may be related to intestinal absorption and utilization of nutrients or digestive enzyme activities. However, how the mechanisms CYP regulate gene expression need to be further studied.

During muscle development, *MYOD1* is a primary myogenic regulator, which is expressed before differentiation of myoblasts [34], and *MYOG* is a secondary myogenic regulator, which is expressed during and after differentiation of myoblasts [35], both of which play a positive regulating role. While *MSTN* is an important negative regulator of muscle tissue growth and negatively regulates muscle growth in animals [36,37]. According to bivariate correlation analysis, *MYOD1*, *MYOG,* and *MSTN* mRNA expression in breast muscle tissues showed no significant correlation with BMP and TMP of the broilers. Moreover, the *MYOD1*, *MYOG,* and *MSTN* mRNA expression in thigh muscle tissues of the CYP_500_ supplemented group was significantly positively correlated with TMP of broilers. Meanwhile, the *MYOG* mRNA expression in thigh muscle tissues of the CYP_1000_ supplementation group was negatively correlated with TMP of broilers. The results demonstrated that diets supplemented with CYP_500_ and CYP_1000_ might affect the growth and development of thigh muscle tissues in broilers. Overall, the *MYOD1*, *MYOG*, and *MSTN* mRNA expression was correlated with TMP of broilers in thigh muscle tissues. Additionally, dietary appropriate CYP supplementation may affect muscle tissue development of broilers by regulating the expression of *MYOD1*, *MYOG*, and *MSTN* mRNA. However, the exact molecular mechanism needs further study.

## 5. Conclusions

In this study, the mRNA expression of *MYOD1*, *MYOG*, and *MSTN* in thigh muscle tissue was significantly positively correlated with TMP in the CYP_500_ supplemented group. *MYOG* mRNA expression in thigh muscle tissues of the dietary CYP_1000_ supplementation group was significantly negatively correlated with TMP. At the same time, adding appropriate CYP to the diet can increase the expression of *MYOD1* and *MYOG* and reduce the expression of *MSTN* mRNA, thereby improving the breast and thigh muscles yield in broilers. Therefore, CYP could be used as a potential feed additive to promote the development of muscle tissues in broilers.

## Figures and Tables

**Figure 1 genes-14-00006-f001:**
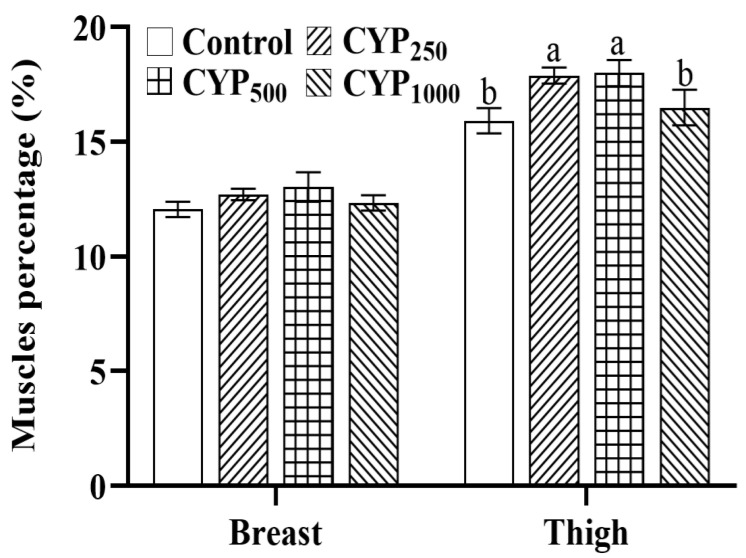
Effects of dietary CYP supplementation on BMP and TMP in broilers. Different lowercase superscripts indicate significant differences (*p* < 0.05).

**Figure 2 genes-14-00006-f002:**
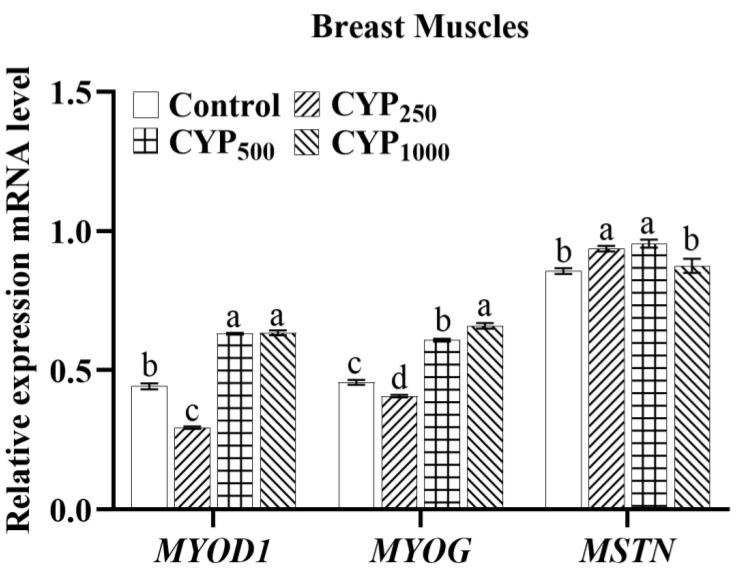
The *MYOD1*, *MYOG*, and *MSTN* mRNA expressions of breast muscle in broilers. Different lowercase superscripts indicate significant differences (*p* < 0.05). *MYOD1*, myogenic differentiation 1; *MYOG*, myogenin; and *MSTN*, myostatin.

**Figure 3 genes-14-00006-f003:**
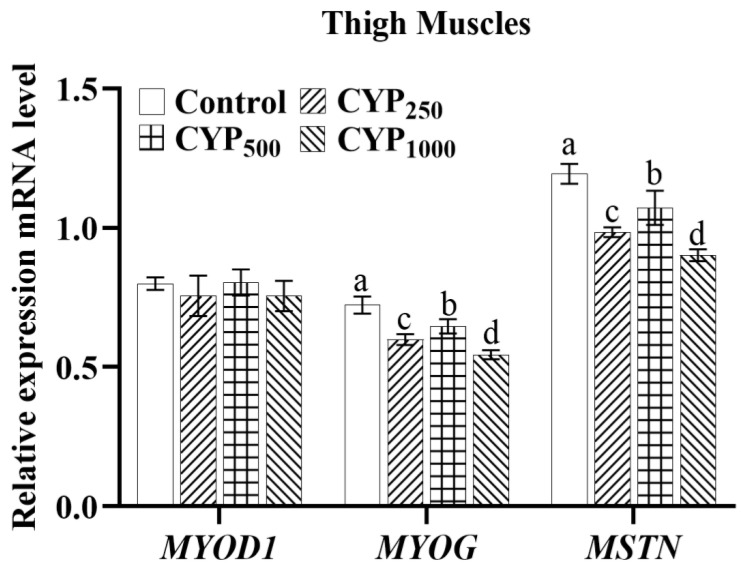
The *MYOD1*, *MYOG*, and *MSTN* mRNA expressions of thigh muscle in broilers. Different lowercase superscripts indicate significant differences (*p* < 0.05). *MYOD1*, myogenic differentiation 1; *MYOG*, myogenin; and *MSTN*, myostatin.

**Table 1 genes-14-00006-t001:** Ingredient and nutrient levels of the basal diet in each feeding phase for broilers.

Items	Composition/%	
	Initial Stages (1–28 days)	End Stages (29–48 days)
Ingredients, %	-	-
Corn	60.00	63.50
Soybean meal	32.00	29.00
Wheat bran	1.00	-
Soybean oil	1.00	2.00
Fish meal	2.00	1.60
CaHPO_4_	1.30	1.30
Limestone	1.40	1.30
NaCl	0.30	0.30
Premix ^a^	1.00	1.00
Total	100.00	100.00
Nutrient levels, %	-	-
Metabolic energy, (MJ/kg) ^b^	12.13	12.55
Crude protein	21.00	20.00
Calcium	1.00	0.90
Total *p*	0.65	0.60
Available *p*	0.45	0.35
Lysine	0.50	0.38
Methionine	1.10	1.00

^a^ Premix supplied per kg: VA 3000 IU, VD_3_ 500 IU, VE 10 IU, VK_3_ 0.5 mg, VB_6_ 3.5 mg, VB_1_ 3.8 mg, D-pantothenic acid 10 mg, folic acid 0.5 mg, biotin 0.15 mg, Fe 80 mg, Cu 8 mg, Zn 75 mg, Mn 60 mg, Se 0.15 mg. ^b^ Metabolic energy was calculated by value and others were measured values.

**Table 2 genes-14-00006-t002:** Primer sequences for RT-qPCR in this experiment.

Gene ^1^	Primer Sequences (5′ → 3′)	Product Size (bp)	GenBank
*β-actin*	Forward: CATTGAACACGGTATTGTCACCAACTG	270	L08165.1
	Reverse: GTAACACCATCACCAGAGTCCATCAC		
*MYOD1*	Forward: ACACGTCGGACATGCACTTCTTC	213	NM_204214
	Reverse: CAGCGTTGGTGGTCTTCCTCTTG		
*MYOG*	Forward: GCGGAGGCTGAAGAAGGTGAAC	347	NM_204184
	Reverse: CGATGGAGGAGAGCGAGTGGAG		
*MSTN*	Forward: TGGCTCTGGATGGCAGTAGTCAG	290	AF019621
	Reverse: CGTCTCGGTTGTGGCATGATAGTC		

^1^ Abbreviations: *MYOD1*, myogenic differentiation 1; *MYOG*, myogenin; and *MSTN*, myostatin.

**Table 3 genes-14-00006-t003:** The correlation between *MYOD1*, *MYOG*, and *MSTN* mRNA expression of breast muscle and BMP or TMP.

Item	Control	CYP_250_	CYP_500_	CYP_1000_
*MYOD1*				
BMP	0.492 (*p* = 0.582)	0.287 (*p* = 0.582)	0.629 (*p* = 0.181)	−0.250 (*p* = 0.633)
TMP	−0.624 (*p* = 0.186)	−0.326 (*p* = 0.528)	−0.625 (*p* = 0.184)	−0.412 (*p* = 0.417)
*MYOG*				
BMP	0.209 (*p* = 0.692)	−0.059 (*p* = 0.912)	−0.298 (*p* = 0.566)	0.421 (*p* = 0.406)
TMP	0.577 (*p* = 0.231)	0.240 (*p* = 0.648)	−0.557 (*p* = 0.251)	0.448 (*p* = 0.373)
*MSTN*				
BMP	0.660 (*p* = 0.154)	0.149 (*p* = 0.779)	−0.461 (*p* = 0.357)	−0.283 (*p* = 0.587)
TMP	−0.348 (*p* = 0.499)	0.128 (*p* = 0.809)	−0.645 (*p* = 0.166)	−0.669 (*p* = 0.147)

BMP, breast muscle percentage. TMP, thigh muscle percentage. *MYOD1*, myogenic differentiation 1; *MYOG*, myogenin; *MSTN*, myostatin.

**Table 4 genes-14-00006-t004:** The correlation between *MYOD1*, *MYOG*, and *MSTN* mRNA expression of thigh muscle and BMP or TMP.

Item	Control	CYP_250_	CYP_500_	CYP_1000_
*MYOD1*				
BMP	0.157 (*p* = 0.767)	0.546 (*p* = 0.263)	−0.228 (*p* = 0.664)	−0.222 (*p* = 0.672)
TMP	0.532 (*p* = 0.277)	0.159 (*p* = 0.764)	0.819 * (*p* = 0.046)	−0.805 (*p* = 0.053)
*MYOG*				
BMP	−0110 (*p* = 0.835)	0.042 (*p* = 0.938)	0.013 (*p* = 0.980)	−0.264 (*p* = 0.614)
TMP	0.585 (*p* = 0.223)	0.291 (*p* = 0.576)	0.912 * (*p* = 0.011)	−0.973 ** (*p* = 0.001)
*MSTN*				
BMP	0.006 (*p* = 0.991)	0.742 (*p* = 0.091)	0.121 (*p* = 0.819)	0.549 (*p* = 0.260)
TMP	0.584 (*p* = 0.223)	0.306 (*p* = 0.556)	0.851 * (*p* = 0.032)	−0.632 (*p* = 0.178)

BMP, breast muscle percentage. TMP, thigh muscle percentage. *MYOD1*, myogenic differentiation 1; *MYOG*, myogenin; *MSTN*, myostatin. A single asterisk * indicates a significant difference (*p* < 0.05), and two asterisks * indicate a very significant difference (*p* < 0.01).

## Data Availability

The data presented in this study are available on request from the corresponding author. The data are not publicly available due to privacy.

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
