# Peer review of "Effects of Chinese yam Polysaccharides on the Muscle Tissues Development-Related Genes Expression in Breast and Thigh Muscle of Broilers"

_genes, 2022, doi:10.3390/genes14010006_

Round 1

Reviewer 1 Report

Journal: Genes

Manuscript ID: genes-2060756

Title: Effects of Chinese Yam polysaccharides on the muscle tissues development-related genes expression in breast and thigh muscle of broilers

Authors: Jiahua Deng, Jinzhou Zhang, Yan Jin, Yadi Chang, Mingyan Shi and Zhiguo Miao

General Comments

1.      Rephrasing of first paragraph of Introduction, so it becomes more comprehendible.

2.      Improve the English of the whole manuscript. 

3.      In Introduction the feed additive investigated must be supported by relevant biological functions. Or the compositional similarities between referenced feed additives must be added with the investigated feed additive, so the biological characteristics of referenced feed additives could be used for writing the Introduction.

4.      Although authors have clearly described the importance of gene investigated in the Introduction of the manuscript but has have not stated how Chinese Yam polysaccharides might be able to effect the expression of muscle related genes. If its absorbed through intestine and regulate muscle development in muscle tissue or is the gene related effect is an indirect consequence of intestinal micro-biome stability. If no data available then authors must consult already published data on feed additives with proprieties similar to CYM and reported to regulate muscle gene.

5.      The abbreviations appearing for the first time in manuscript must be written in full and abbreviation in parenthesis and then subsequently can be written in abbreviated form, especially in Material and Method.

6.      Why he trial was divided in two feeding phases day1-28 and day29-48. As three-phase feeding regime is followed world-wide. So authors are requested to explain the division for clarity for readers.

7.      The main theme of study is CYP increase skeletal muscle yield and the basic mechanism behind this muscle development is controlled by expression of different genes. Si I would suggest instead of stating that CYP regulates different genes which led to muscle development, consider CYP led improvement in gut, leading to higher bioavailability of nutrients and subsequent muscle development evident by higher gene expression.  

8.      Discussion lack the supporting point for achieving the main objective related to CYM gene expression “

Introduction:

1.      Line 35: “Chinese herbal medicine is a kind green feed additive”. As Chinese Yam polysaccharides is an extracted product as mentioned in subsequent lines, so I would suggest adding “Chinese herbs are a source of green feed additive”.

2.      Line 39: It has a variety of biological” instead write “they have”.

3.      Line 39: “It has a variety of biological functions, such as regulating intestinal microflora, anti-oxidant and anti-inflammatory responses anti-tumor, and so on” There must be a proper description for biological functions of Chinese Yam polysaccharides, instead the reference that you added with wording indicating the biological functions of Chinese Yam polysaccharides are for polysaccharides from purple sweet potato, Chlorella is a single-celled, non-motile green alga and chitooligosaccharide. They are not Chinese Yam”.

Material and Method:

1.      Line 81: (1 D) must be replaced with day-old.

2.      Line 90: with more than carbohydrate level of 90.40% must be replaced with “carbohydrate level more than 90%”.

3.      Line 90-91: Please rephrase it for better understanding as per previous comment.

4.      Please explain the percentages written in the line: 91. What mean by polysaccharide content of 30.00% among them, monosaccharide types?

5.      The expression analysis should be described with more detail.

Results:

            Satisfactory

Discussion:

Line 205-06:  CYP could improve the muscle tissue development of broilers by regulating the expression of muscle related genes. Please explain the exact mechanism CYP regulate gene expression.

Line 212-220: The repetition of result description without any discussion point.

Author Response

Response to Reviewer 1 Comments

Dear Reviewers:

Thank you for your careful review and constructive suggestions regarding our manuscript. We have revised the manuscript in accordance with the comments and marked all the amends on our revised manuscript.

Point 1: Rephrasing of first paragraph of Introduction, so it becomes more comprehendible.

Response 1: Thank you very much for your advice. We have revised the wording of the first paragraph of the introduction, in the yellow part of the first paragraph.

Point 2: Improve the English of the whole manuscript.

Response 2: Thank you very much for your advice. We thank the reviewer for this valuable suggestion, and the whole manuscript has been polished accordingly.

Point 3: In Introduction the feed additive investigated must be supported by relevant biological functions. Or the compositional similarities between referenced feed additives must be added with the investigated feed additive, so the biological characteristics of referenced feed additives could be used for writing the Introduction.

Response 3: Thank you very much for your suggestion. Our previous research found that CYP has a positive effect on the growth performance and immune function of crossbred broilers, but there is no research report on the effect of CYP on the muscle development of crossbred broilers. Therefore, this experiment is carried out under this background.

Point 4: Although authors have clearly described the importance of gene investigated in the Introduction of the manuscript but has have not stated how Chinese Yam polysaccharides might be able to effect the expression of muscle related genes. If its absorbed through intestine and regulate muscle development in muscle tissue or is the gene related effect is an indirect consequence of intestinal micro-biome stability. If no data available then authors must consult already published data on feed additives with proprieties similar to CYP and reported to regulate muscle gene.

Response 4: Thank you very much for your advice. CYP maybe through intestine and regulate muscle development in muscle tissue or is the gene related effect is an indirect consequence of intestinal micro-biome stability, however, there is still no relevant evidence. Because CYP is the research hotspot of feed additives in recent years, no research report has been found on feed additives with similar characteristics to CYP.

Point 5: The abbreviations appearing for the first time in manuscript must be written in full and abbreviation in parenthesis and then subsequently can be written in abbreviated form, especially in Material and Method.

Response 5: Thank you very much for your advice. We have already checked and followed this rule.

Point 6: Why he trial was divided in two feeding phases day1-28 and day29-48. As three-phase feeding regime is followed world-wide. So authors are requested to explain the division for clarity for readers.

Response 6: Thank you very much for your suggestion. This trial is divided into two feeding phases day1-28 and day29-48, main considers the addition cost of broiler feeding. In Journal Frontiers in Veterinary Science, our previous research report the experiment of hybrid broilers into two feeding trial was divided in two feeding phases day1-28 and day29-48.

Point 7: The main theme of study is CYP increase skeletal muscle yield and the basic mechanism behind this muscle development is controlled by expression of different genes. Si I would suggest instead of stating that CYP regulates different genes which led to muscle development, consider CYP led improvement in gut, leading to higher bioavailability of nutrients and subsequent muscle development evident by higher gene expression.

Response 7: Thank you very much for your suggestion. We have done the experiments on the CYP improve the intestinal morphology of hybrid broilers, but these data have been used on another manuscript (submit earlier than this manuscript, and manuscript entitled ‘Effects of dietary Chinese Yam Polysaccharide on growth performance, small intestinal morphology and digestive enzyme activities of broilers’) which now is still under review. According your suggestion, if possible, in the future studies we may further consider to investigate CYP led improvement in gut, leading to higher bioavailability of nutrients and subsequent muscle development evident by higher gene expression of broilers.

Point 8: Line 35: “Chinese herbal medicine is a kind green feed additive”. As Chinese Yam polysaccharides is an extracted product as mentioned in subsequent lines, so I would suggest adding “Chinese herbs are a source of green feed additive”.

Response 8: Thank you very much for your suggestion. Thank you very much for your advice. We have revised the mistake, “Chinese herbal medicine is a kind green feed additive” in line 34-35 has been revised as “Chinese herbs are a source of green feed additive” (In red: Line 34-35).

Point 9: Line 39: It has a variety of biological” instead write “they have”.

Response 9: Thank you very much for your advice. We have revised the mistake, “It has” in line 38-39 has been revised as “they have”. (In red: Line 38-39).

Point 10: Line 39: “It has a variety of biological functions, such as regulating intestinal microflora, anti-oxidant and anti-inflammatory responses anti-tumor, and so on” There must be a proper description for biological functions of Chinese Yam polysaccharides, instead the reference that you added with wording indicating the biological functions of Chinese Yam polysaccharides are for polysaccharides from purple sweet potato, Chlorella is a single-celled, non-motile green alga and chitooligosaccharide. They are not Chinese Yam”.

Response 10: Thank you for your question. In this sentence, “It” refers to the Chinese herbal polysaccharides, not specifically the Chinese Yam polysaccharides. Therefore, the described biological functions is the biological functions of several Chinese herbal polysaccharides, not just Chinese Yam polysaccharides. In addition, purple sweet potato and chlorella are included in the Pharmacopoeia of the People’s Republic of China as Chinese herbal medicine. We have revised the mistake, “It has” in line 38-39 has been revised as “they have”. (In red: Line 38-39).

Point 11: Line 81: (1 D) must be replaced with day-old.

Response 11: Thank you very much for your suggestion. We have revised the mistake, “1 D” in line 80 has been revised as “1-day-old” (In red: Line 80).

Point 12: Line 90: with more than carbohydrate level of 90.40% must be replaced with “carbohydrate level more than 90%”.

Response 12: Thank you very much for your suggestion. We have revised the mistake, “with more than carbohydrate level of 90.40%” in line 91 has been revised as “carbohydrate level more than 90%” (In red: Line 91).

Point 13: Line 90-91: Please rephrase it for better understanding as per previous comment.

Response 13: Thank you very much for your suggestion. We have rephrased in the line 96-97 (In red: Line 91).

Point 14: Please explain the percentages written in the line: 91. What mean by polysaccharide content of 30.00% among them, monosaccharide types?

Response 14: Thank you for your question. The content of polysaccharide in the yam polysaccharide we purchased from the Shanxi Hana Biotechnology Co., Ltd (Shanxi, P. R. China) accounts for 30% of the total bioactive substances.  The yam polysaccharide is mainly composed of two monosaccharides, glucose and galactose, with the content of 99.48% and 0.52% respectively.

Point 15: The expression analysis should be described with more detail.

Response 15: Thank you for your question. This trial is a part of the Program for Innovative Research Team (in Science and Technology) in University of Henan Province (22IRTSTHN026). Before the end of the project, more detailed expression analysis cannot be provided temporarily.

Point 16: Line 205-06:  CYP could improve the muscle tissue development of broilers by regulating the expression of muscle related genes. Please explain the exact mechanism CYP regulate gene expression.

Response 16: Thank you very much for your suggestion. At present, we do not know the underlying mechanism CYP regulate gene expression, so in the Discussion section we speculate the possible reasons. We have added “All of these results suggest that the CYP promotes broiler muscle development by regulating the expression of muscle related genes, which may be related to intestinal absorption and utilization of nutrients or digestive enzyme activities” in the line 200-204. (In red: Line 200-204).

Point 17: Line 212-220: The repetition of result description without any discussion point.

Response 17: Thank you very much for your suggestion. We have deleted the repeated result discussion and added the corresponding discussion points in the line 200-204. (In red: Line 200-204).

In addition:

The repetition rate of the manuscript proposed by the editor is a little high. We have already modified and marked with yellow highlight.

Sincerely,

Zhiguo Miao

Reviewer 2 Report

I have significant reservations about the experimental design itself. Was pen or bird the experimental unit? You state that there is 3 replicates (Line 15 & 83), yet you selected 6 birds at random from each group (Line 99). If pen is your experimental unit then I worry if 3 replicates is enough for appropriate statistical analysis. If bird is the experimental unit, then it would need to be clarified in the manuscript. 

Typically muscle percentages are calculated as a percentage of live body weight and referred to as yields.  

The formatting of the manuscript needs to be addressed. There is alignment issues with the text, tables/figures, table headers/footings, and references. Be careful not to have headings or subheadings as the final line on a page. I would also consider combining the word "muscle" in your title. I understand why it's like that, but I foresee confusion when peers reference your manuscript in the future. 

Author Response

Response to Reviewer 2 Comments

Dear Reviewers:

Thank you for your careful review and constructive suggestions regarding our manuscript. We have revised the manuscript in accordance with the comments and marked all the amends on our revised manuscript.

Point 1: I have significant reservations about the experimental design itself. Was pen or bird the experimental unit? You state that there is 3 replicates (Line 15 & 83), yet you selected 6 birds at random from each group (Line 99). If pen is your experimental unit then I worry if 3 replicates is enough for appropriate statistical analysis. If bird is the experimental unit, then it would need to be clarified in the manuscript.

Response 1: Thank you very much for your suggestion. In this study, replicate was used as experimental unit for statistical analysis and “Replicate used as experimental materials unit was analyzed individually for the study of each experimental parameter” has been inserted in line 101. (In red: Line 101).

Point 2: Typically muscle percentages are calculated as a percentage of live body weight and referred to as yields.

Response 2: Thank you very much for your suggestion. The muscle percentage of this test is calculated by reference to ‘Effects of integrated rice-duck farming system on duck carcass traits, meat quality, amino acid, and fatty acid composition’. The study was published in the Journal of Poultry Science.

Point 3: The formatting of the manuscript needs to be addressed. There is alignment issues with the text, tables/figures, table headers/footings, and references. Be careful not to have headings or subheadings as the final line on a page. I would also consider combining the word "muscle" in your title. I understand why it's like that, but I foresee confusion when peers reference your manuscript in the future.

Response 3: Thank you very much for your suggestion. The format of the manuscript has been modified according to the requirements of this journal before submission. I think the format problem you raised may be related to the software version of the manuscript. In our manuscript, there is no title or subtitle as the last line of the page. Adding the word "muscle" to the title is based on the perception of the title of the latest report in this direction.

In addition:

The repetition rate of the manuscript proposed by the editor is a little high. We have already modified and marked with yellow highlight.

Sincerely,

Zhiguo Miao

Reviewer 3 Report

Authors did a good job. Few additions would help to provide more clarity.

1.      360 growing broilersshould be changed to 360 broiler chicks” (line 14 and 81).

2.      Author should mention the sex of broiler chicks (male, female or mixed sex) and breed (line 14 and 81).

3.      How to get the Nutrient levels in Table 1?  Calculated versus Analytical Nutrient Values.

4.      Adding Initial stages (1-28 D)” and End stages (29-48 D)” into Table 1.

5.      Adding more detail on animal care and management in “2.1. Experimental Design, Diets and Broilers”

6.      All superscripts of BMP (Figure 1) and MYOD1 mRNA expressions (Figure 3) should be deleted because there was no significant.

7.      Author should not say “promote the growth” because this experiment did not determine these parameters (line 29-30 and 233-234).

Author Response

Response to Reviewer 3 Comments

Dear Reviewers:

Thank you for your careful review and constructive suggestions regarding our manuscript. We have revised the manuscript in accordance with the comments and marked all the amends on our revised manuscript.

Point 1: “360 growing broilers” should be changed to “360 broiler chicks” (line 14 and 81).

Response 1: Thank you very much for your suggestion. We have revised the mistake, “360 growing broilers” in line 13-14 and 80-81 has been revised as “360 broiler chicks” (In red: Line 13-14 and 80-81).

Point 2: Author should mention the sex of broiler chicks (male, female or mixed sex) and breed (line 14 and 84).

Response 2: Thank you very much for your suggestion. We have added “sex of broiler chicks (male, female or mixed sex) and breed” in the line 13-14 and 80-81 (In red: Line 13-14 and 80-81).

Point 3: How to get the Nutrient levels in Table 1? Calculated versus Analytical Nutrient Values.

Response 3: Thank you very much for your suggestion. The nutrition level in Table 1 is calculated based on the poultry nutrition requirements formulated by the National Research Council of the United States and the nutrient composition of each raw material. Metabolic energy was calculated by value and others were measured values.

Point 4: A Adding “Initial stages (1-28 D)” and “End stages (29-48 D)” into Table 1.

Response 4: Thank you very much for your suggestion. We have added “Initial stages (1-28 D)” and “End stages (29-48 D)” in the Table 1 (In red: Table 1).

Point 5: Adding more detail on animal care and management in “2.1. Experimental Design, Diets and Broilers”.

Response 5: Thank you very much for your advice. We have added more detail on animal care and management in the line 88-90 (In red: Line 88-90).

Point 6: All superscripts of BMP (Figure 1) and MYOD1 mRNA expressions (Figure 3) should be deleted because there was no significant.

Response 6: Thank you very much for your advice. We have modified Figure 1 and Figure 3” (In red: Figure 1 and Figure 3).

Point 7: Author should not say “promote the growth” because this experiment did not determine these parameters (line 29-30 and 233-234).

Response 7: We have revised the mistake, “promote the growth” in line 28-29 and 231-232 has been revised as “promote the development of muscle tissues” (In red: Line 28-29 and 231-232).

In addition:

The repetition rate of the manuscript proposed by the editor is a little high. We have already modified and marked with yellow highlight.

Sincerely,

Zhiguo Miao

Round 2

Reviewer 1 Report

Line 38: Scientifc name should be itlaicized. 

Line 48-49: The paper that authors are refering to as reference is describing the linkage of monosacchrides, so improve the line 48 and 49 accodring to the comment response you made as "the yam polysaccharide is mainly composed of two monosaccharides, glucose and galactose, with the content of 99.48% and 0.52% respectively."

Response 3: Thank you very much for your suggestion. Our previous research found that CYP has a positive effect on the growth performance and immune function of crossbred broilers, but there is no research report on the effect of CYP on the muscle development of crossbred broilers. Therefore, this experiment is carried out under this background.

Comment: I could not find any paper "referred to" in Introduction reagdung CYP supplementation in "crossbred broilers". Instead the authors are referring to paper of rat where diet was supplemented with CYP. 

Author Response

Response to Reviewer 1 Comments

Dear Reviewers:

Thank you for your careful review and constructive suggestions regarding our manuscript. We have revised the manuscript in accordance with the comments and marked all the amends on our revised manuscript.

Point 1: Scientific name should be italicized.

Response 1: Thank you very much for your advice. We have changed the scientific name to italic format. (In red: Line 36-37)

Point 2: The paper that authors are referring to as reference is describing the linkage of monosaccharides, so improve the line 48 and 49 according to the comment response you made as "the yam polysaccharide is mainly composed of two monosaccharides, glucose and galactose, with the content of 99.48% and 0.52% respectively."

Response 2: Thank you very much for your advice. Our reference [11] describes the relationship between monosaccharides, which means that the CYP in reference [11] are mainly composed of glucose and galactose. However, in lines 91-92, we describe that the CYP is mainly composed of 99.48% glucose and 0.52% galactose, meaning that the CYP used in this study was detected to be composed of 99.48% glucose and 0.52% galactose. This shows that the main ingredients of CYP we used are similar to those in references [11], which are composed of glucose and galactose.

Point 3: Response 3: Thank you very much for your suggestion. Our previous research found that CYP has a positive effect on the growth performance and immune function of crossbred broilers, but there is no research report on the effect of CYP on the muscle development of crossbred broilers. Therefore, this experiment is carried out under this background.

Comment: I could not find any paper "referred to" in Introduction regarding CYP supplementation in "crossbred broilers". Instead the authors are referring to paper of rat where diet was supplemented with CYP.

Response 3: Thank you very much for your advice.  We have added in lines 54-56, “Our previous study indicated also confirmed that CYP has a positive effect on improving immune function of crossbred broilers”.  (In red: Line 54-56). In addition, we have done the experiments on the CYP improve the growth performance of hybrid broilers, but these data have been used on another manuscript (submit earlier than this manuscript, and manuscript entitled ‘Effects of dietary Chinese Yam Polysaccharide on growth performance, small intestinal morphology and digestive enzyme activities of broilers’) which now is still under review.

In addition:

We have changed the scientific name to italic format. (In red: Line 2, 10, 46, 181)

现在,我们同意将张晋洲从共同第一作者中删除,只保留他的作者身份。

真诚地

苗志国
